# Evaluation of Agrobiodiversity and Cover Crop Adoption in Southern Ontario Field Crops

Katherine Shirriff, Krishna Bahadur KC *⬤ and Aaron Berg ⬤

Department of Geography, Environment and Geomatics, University of Guelph, Guelph, ON N1G 2W1, Canada; katherineshirriff20@gmail.com (K.S.); aberg@uoguelph.ca (A.B.)
* Correspondence: krishnak@uoguelph.ca; Tel.: +1-519-824-4120 (ext. 58189)

**Abstract:** Incorporating cover crops into corn and soybean operations across Southern Ontario is essential for maintaining yields under environmental stressors. Unfortunately, amongst the literature, there is a concern about the low adoption rate of cover crops in the northern Corn Belt due to a shift toward low agrobiodiversity and dominance of more profitable corn and soybean cropping systems, encouraged by extensive use of fertilizers, herbicides, and pesticides. This study examines whether Southern Ontario is following suit in decreasing agrobiodiversity trends, at the county level, and examines the adoption of cover crops within corn and soybean operations across Southern Ontario using digital imagery from 2013 to 2018. Results reveal that Southern Ontario is indeed shifting from systems characterized by higher agrobiodiversity to systems dominated with corn, soybean, and hay. Despite the benefits of cover crops, this study reveals that most of the current corn and soybean operations are not incorporating cover crops into the rotation. More significantly, the low adoption of cover crops is most apparent in southwestern Ontario, and increases in adoption occur toward the north.

**Keywords:** agrobiodiversity; climate change; cover crops; Ontario

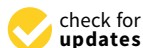



## 1. Introduction

Climate change has the potential to challenge global production of a sustainable food supply. Climate change promotes inclement climatic conditions, including extreme weather patterns, such as heatwaves, prolonged dry spells, increased rainfall, hailstorms, and other adverse weather conditions. Northern agroecosystems have been challenged by unfavorable climatic conditions, as reflected in a 13% reduction of corn and soybean production in North America and Europe from decreased rainfall and excessive heat [1]. North America and Europe produce approximately 45% and 32% of the worldwide supply of corn, wheat, and soybean, respectively [2], making the global supply of corn and soybean vulnerable to a changing climate. Southern Ontario, located in the northern Corn Belt, is a significant area of concern for corn and soybean loss due to climate change, as it produces approximately 50 and 60% of Canada's corn and soybean, respectively [3]. More significantly, various climate models suggest that summer and winter in Ontario will experience an increase in temperature with more frequent extreme weather events, dry spells, heat waves, and intense rainfall—conditions that are unfavorable to Ontario's crops [4,5]. To secure corn and soybean production under more challenging environmental conditions in Southern Ontario, there is a need to adopt effective and sustainable cropping systems to maintain yields.

Cropping sequence diversification, including the use of cover crops, is an attractive solution to improve field crop resilience to environmental stresses. Agroecosystem diversity is strongly advocated for its utility in improving the resilience and stability of cropping systems under stressed conditions [6,7]. Cover crops are commonly defined as plants that cover the soil and are primarily used to secure soil functionality by controlling soil erosion

and nitrogen leaching [8–13]. Winter annual cover crops were regularly incorporated into many North American cropping systems during the early 20th century [14]. However, following World War II, the use of cover crops for soil conservation declined due to the emergence of synthetic fertilizers, herbicides, and pesticides. Research by Singer et al. [15] suggests that the northern Corn Belt is following the practices observed in the southern limits of this region, including low adoption rates of cover crops. The importance of cover crop systems has re-emerged as a necessary soil conservation technique in response to the global recognition of soil degradation [16]. Cover crops should be incorporated into corn and soybean operations in Southern Ontario cropping systems to add diversity and resilience to the cropping systems and maintain yields under stressed conditions.

There is ample research that describes the effects of cover crops on corn and soybean yields and trends in its adoption [7,9,16–19]; however, the extent to which corn and soybean operations are incorporating cover crops into short corn/soybean rotations across Southern Ontario has yet to be assessed. Research by Singer et al. [15] provided a useful exploration in quantifying cover crop adoption trends across the northern Corn Belt, utilizing mail-in surveys. Other studies have discussed trends in cover crop adoption using satellite imagery in northern United States [20,21]. There is extensive research that quantifies cover crop adoption rates in northern United States; however, the locations of these studies have not targeted Southern Ontario. More significantly, the extent to which cover crops are adopted in corn and soybean operations, as well as trends in cover crop spatial adoption across Southern Ontario, are yet to be explored.

This study investigates trends in agricultural diversification and the adoption of cover crops in corn–soybean rotations using satellite-derived crop surveys of Southern Ontario, Canada. This research consists of two analyses. The first examines whether Southern Ontario is decreasing in agricultural crop diversity, toward dominance in corn and soybean, by identifying trends in field crop production using the Ontario Ministry of Agriculture, Food and Rural Affairs' (OMAFRA) [22] Field Crop data from 1981 to 2018. More significantly, the field crop trends are investigated on the county scale to identify spatial trends in crop diversity. Following this, the second goal of this study is to investigate the adoption of cover crops within corn and soybean cropping systems across Southern Ontario at high spatial resolutions. Using Agriculture and Agri-Food Canada (AAFC) annual crop inventory (ACI) dataset [23] compatible with Esri ArcMap, this study analyzes continuous corn, continuous soybean, corn–soybean rotations, corn–cover crop rotations, soybean–cover crop rotations, and corn–soybean–cover crop rotations at a 30 m resolution to assess spatial trends in agrobiodiversity and cover crop adoption in corn and soybean cropping systems. A variety of cover crops are utilized in Southern Ontario; however, rye, oats, and winter wheat are the most commonly adopted cover crops in Southern Ontario and tracked by the ACI dataset. Several other commonly adopted cover crops, such as red clover, typically planted at the end of the growing season or inter-seeded into actively growing crops, are commonly used in Ontario. However, due to some of the challenges in obtaining accurate and timely ground cover information during the non-growing season, including difficulties with high cloud cover and seasonal snow coverage [24,25], this study limits the focus only to cover crops that are tracked in the ACI during the following growing season, including rye, oats, and winter wheat data (in this case each of these potential cover crops were planted in the fall of the preceding year). The objective of this part of the analysis is to explore the frequency with which corn and soybean productions have incorporated one of these cover crop mixes within the six years of the study period.

## 2. Methods

### 2.1. Study Area

Agriculture in Ontario accounts for 25.6% of agricultural production in Canada [3]. Agricultural production in Ontario is prominent in Central and Southern Ontario, occupying approximately 4.1 million acres, or 44.2% of the total area [26]. Prime agricultural soils in this area allow for the region to generate the most significant production of soybean, corn,

and winter wheat in the country, representing 60, 50, and 57% of Canada's total production of each crop, respectively [3]. On the provincial scale, oilseed and grain type operations (including corn, soybean, and winter wheat) account for 34% of Ontario's agricultural production [3]. Following oilseed and grain, other crops and beef production are the second and third largest production operations in Ontario [3]. A focus on field crop operations across Southern Ontario is analyzed in this study to investigate agrobiodiversity trends, as well as the adoption of corn–soybean–winter wheat rotations (Figure 1).

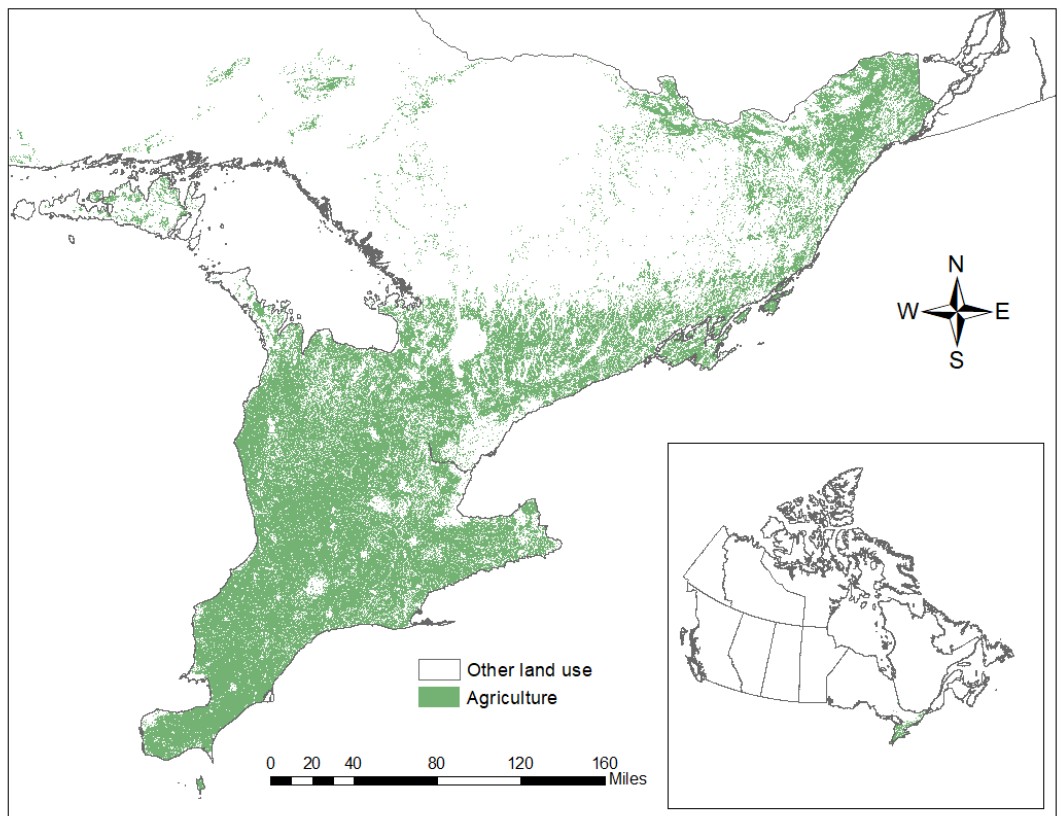

**Figure 1.** Areas of field crop production in Southern Ontario. Image manipulated from Agriculture and Agri-Food Canada (AAFC) [23] Annual Crop Inventory (ACI).

### 2.2. Identifying Trends in Field Crop Production and Diversification across Southern Ontario

This research utilizes Field Crop Data provided by OMAFRA [22] from 1981 to 2018 to investigate the diversification of field crops across Southern Ontario. The Field Crop dataset contains a provincial estimate of harvested area for winter wheat, spring wheat, buckwheat, rye, oats, barley, mixed grain, grain corn, canola, soybeans, flaxseed, dry white beans, beans, fodder corn, and hay for each year. From this dataset, five categories were established for analysis, including corn, soybean, winter wheat, hay, and other field crops (oats, barley, mixed grain, white beans, spring wheat, canola, flaxseed). Temporal trends of corn, soybean, winter wheat, hay, and other field crops are presented using a percent stacked area graph.

OMAFRA's Area and Production Estimates by County (2004–2016) dataset was utilized to identify spatial field crop trends at the county scale. Similar to the Field Crop dataset, this dataset consists of production estimates measured in hectares of counties. The included crops in the Area and Production Estimates by County (2004–2016) dataset excludes rye, flaxseed, and considers colored beans—an inconsistency from the previously utilized Field Crop Data. The field crops in this dataset were also grouped into five categories: corn, soybean, winter wheat, hay, and other field crops (oats, barley, white beans, spring wheat, canola, colored beans). A Kendall-tau b two-tailed test (Equation (1)) was used to determine significant trends of each field crop group. This test identifies counties that are significantly

increasing or decreasing the production of certain field crops. The Kendall-tau b test is defined as:

$$Tb = (nc - nd)\sqrt{((n0 - n1) \times (n0 - n2))} \tag{1}$$

where:

$n0 = n(n - 1)/2$
$n1 = \Sigma_i\, t_i(t_i - 1)/2$
$n2 = \Sigma_j\, t_j(t_j - 1)/2$
$nc$ = Number of concordant pairs
$nd$ = Number of discordant pairs
$t_i$ = Number of tied values in the ith group of ties for the first quantity
$t_j$ = Number of tied values in the jth group of ties for the second quantity

Correlation strength is computed for *p*-values, R2, and r-values for each field crop group. Trends were mapped using ArcMap. Increasing and decreasing trends are identified for each county using a blue and red gradient representing decreasing and increasing trends, respectively. An "x" is used as an identifier on counties that contain a significant trend.

*2.3. Analysis of Winter Wheat, Rye, and Oat Cover Crop Adoption in Corn and Soybean Rotations across Southern Ontario*

The ACI dataset provided by AAFC is useful for analyzing the adoption of cover crops within corn and soybean rotations across Southern Ontario. This dataset derives land use classifiers from remote sensing data and delivers a national crop inventory at a 30 m resolution. This study utilizes ACI data to investigate the frequency of which cover crops are adopted into corn and soybean cropping systems between 2013 and 2018. A six-year study period is desirable for this study, as it incorporates two completed corn–soybean cover crop rotations.

Using Esri ArcMap, the study site was extracted from the six yearly datasets, and pixels were reclassified to display corn, soybean, or cover crops (rye, oats, or winter wheat), and other land use was assigned null data. OMAFRA [27] identifies rye, oat, and winter wheat as some common cover crops grown in Southern Ontario. This study acknowledges that rye, oats, and winter wheat are not always grown as cover crops; however, for the purposes of this study, they will be identified as cover crops. The derived datasets were then overlaid, and pixels that are exclusive to corn, soybean, or cover crops throughout the six years were extracted and analyzed. Rotations were classified based on the frequency of occurrence of corn, soybean, or cover crops over six years for every pixel. From this, 28 possible rotations were identified (Table 1). The analyzed rotations exclude productions that have incorporated other crops or cover crops into the rotation during the study period, including pasture, barley, and spring wheat. Given this constraint, the focus is limited to approximately 28% of the agricultural area of Southern Ontario that has produced corn, soybean, rye, oats, or winter wheat in some capacity over the past six years and 17% of all agricultural operations during this time.

A tri-variate color scheme was developed in this study to represent each rotation classification. This method uses a three-color gradient to represent the integration of corn, soybean, and cover crops in rotations (Figure 2 left side). Red, green, and dark blue represent the three vertices of the tri-variate color scheme. The vertices represent pixels where continuous corn (rotation code 1), continuous cover crops (rotation code 22), and continuous soybean (rotation code 28) have occurred over the study period. Rotation code numbers located between two vertices adopt a color that incorporates both vertex colors, representing the presence of both crops in the rotation. For instance, rotation code 10 (6:0:6) represents a pixel that has grown corn for three years and soybean for three years, so it displays a magenta color, representing an equal mix of soybean (blue) and corn (red) (Figure 2 left side). The color gradient becomes less vibrant closer to the centroid, as the code numbers in this location incorporate all three crop colors. Rotation code 13 (2:2:2) represents an equal occurrence of corn, soybean, and cover crops, as each crop has occurred

two times over the six years. Pixels containing rotation code 13 are symbolized as grey, representing a uniform mixing of corn, soybean, and winter wheat.

**Table 1.** Rotation possibilities in Southern Ontario from 2013 to 2018 using the ratio of corn, winter wheat, and soybean grown within a 6-year period (corn: winter wheat: soybean).

| Possible Rotation Type (Rotation Code) | Corn: Cover Crop: Soybean |
|:---:|:---:|
| 1 | 6:0:0 |
| 2 | 5:1:0 |
| 3 | 5:0:1 |
| 4 | 4:2:0 |
| 5 | 4:1:1 |
| 6 | 4:0:2 |
| 7 | 3:3:0 |
| 8 | 3:2:1 |
| 9 | 3:1:2 |
| 10 | 3:0:3 |
| 11 | 2:4:0 |
| 12 | 2:3:1 |
| 13 | 2:2:2 |
| 14 | 2:1:3 |
| 15 | 2:0:4 |
| 16 | 1:5:0 |
| 17 | 1:4:1 |
| 18 | 1:3:2 |
| 19 | 1:2:3 |
| 20 | 1:1:4 |
| 21 | 1:0:5 |
| 22 | 0:6:0 |
| 23 | 0:5:1 |
| 24 | 0:4:2 |
| 25 | 0:3:3 |
| 26 | 0:2:4 |
| 27 | 0:1:5 |
| 28 | 0:0:6 |

A second, more general map groups the 28 rotation types into 7 types containing similar proportions (Table 2), providing a more general description of areas where monoculture, 2-crop rotation, and 3-crop rotation occur over the six-year study period. A similar tri-variate approach as mentioned above is used for this map; however, the symbology is colored based on the rotation type—whether that be a monoculture, two-crop rotation, or three-crop rotation. Red represents rotations that have only grown one or two crops in rotation within six years. Dark red is located at the vertices representing monoculture and lightens between vertices to represent a regular occurrence of crops within a two-crop rotation. Blue represents rotations where all three crops (three-crop rotation) appear within the rotation, and green represents an equal occurrence of all three crops appearing within the 6-year rotation period (Figure 2 right-side).

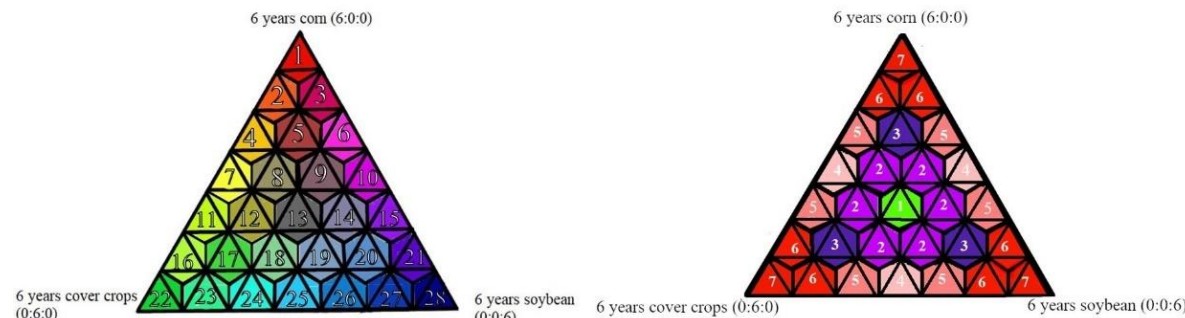

**Figure 2.** Types of rotation in southern Ontario: (**Left**). 28 possible rotation types for corn, winter wheat, and soybean over a 6-year study period (**Right**). 7 possible types, representing the same ratios of the number of crops grown within the six years.

**Table 2.** Types of aggregated rotations based on crop variability and frequency. The aggregated rotations are based on Table 1, showing the frequency of corn: cover crops: soybean grown within a 6-year period.

| Rotation Type | Code | Corn: Cover Crops: Soybean |
|---|---|---|
| 3-crop rotation | 1 | 2:2:2 |
| | 2 | 3:2:1 or: 2:3:1 or 1:2:3 or 1:3:2 or 3:1:2 or 2:1:3 |
| | 3 | 4:1:1 or 1:4:1 or 1:1:4 |
| 2-crop rotation | 4 | 3:3:0 or 0:3:3 or 3:0:3 |
| | 5 | 4:2:0 or 4:2:0 or 0:4:2 or 0:4:2 or 4:0:2 or 4:0:2 |
| | 6 | 5:1:0 or 1:5:0 or 0:5:1 or 0:1:5 or 1:0:5 or 5:0:1 |
| Monoculture | 7 | 6:0:0 or 0:6:0 or 0:0:6 |

## 3. Results and Discussion

### 3.1. Field Crop Trends across Southern Ontario

Across Southern Ontario, crop varieties, and consequently, the likelihood of a more significant occurrence of complex rotations, have reduced due to the reduction in small grains and forage production. Figure 3 reveals a decline in crop diversity from 1981 to 2018, as "other" field crops appear to be declining, while corn, soybean, winter wheat, and hay dominate production. Soybean and winter wheat show an increasing trend. In contrast, corn and hay are declining slightly, and other field crops are decreasing at a more significant trend. Although hay and corn are decreasing, they continue to represent a substantial proportion of field crop production. The observed increase in soybean and the dominance of corn production corresponds with the observations of Wright and Wimberly [28]. Their study addresses the westward migration of beef and, in turn, a decrease in forage and grassland production in this area. Increasing corn and soybean production is concurrent with the westward migration of beef and increasing demands for biofuel [29]. The dominance of corn and soybean production and the decrease in other field crops are concerning, as it suggests that agrobiodiversity is declining.

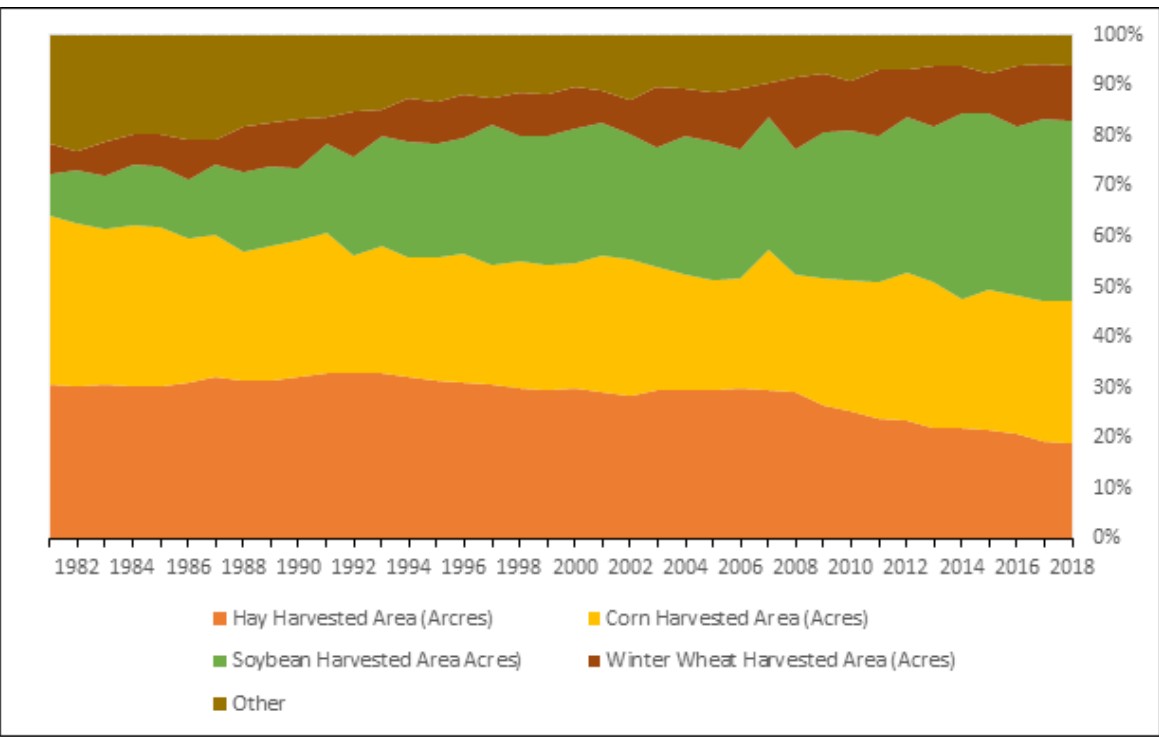

**Figure 3.** Percent of each crop production relative to the total harvested area (acres) in Ontario from 1981 to 2018 [22].

Spatial trends of crop production vary between the five categories. Investigating field crop production trends on a county scale, Figure 4 illustrates the trends of corn, soybean, winter wheat, hay, and other field crop production from 2004 to 2018. Similar to Liebman et al. [29], Figure 4 suggests that corn, soybean, and winter wheat are increasing across most counties in Southern Ontario, while hay and other field crops are decreasing. Figure 4 illustrates a trend of increasing soybean production across counties in Southern Ontario, with 13 statistically significant counties increasing in soybean production, preceding north and northeast in Southern Ontario. In contrast, other field crops and hay appear to be declining across Southern Ontario, with more than half of the counties showing statistically significant decreases in other field crops. This is observable in counties within the southwest and proceeding north and northeast across Southern Ontario. With this considered, other field crops and hay seem to complement soybean production trends. Research by Hume and Pearson [30] supports this observation, suggesting that the eastward migration of soybean in Ontario is a result of earlier-maturing soybean cultivar development. Before 1975, soybean production was limited to southwestern Ontario due to the warm climatic conditions [30]. Due to the development of earlier-maturing soybeans, production expanded to colder regions of Ontario, promoting the north and northeastern migration of soybean production [30]. The increase in soybean in north and northeastern counties resulted in the decline of cereal grains, except wheat, by 2000 [30]. Hume and Pearson's (2011) study is consistent with trends in soybean and other crops from 2004 to 2018 presented in this research. The decline of hay production is prominent amongst many Southern Ontario's counties. This trend is consistent with Gaudin et al., [7,17] study, which suggests the decline of grassland and forage is a result of a reduction of livestock.

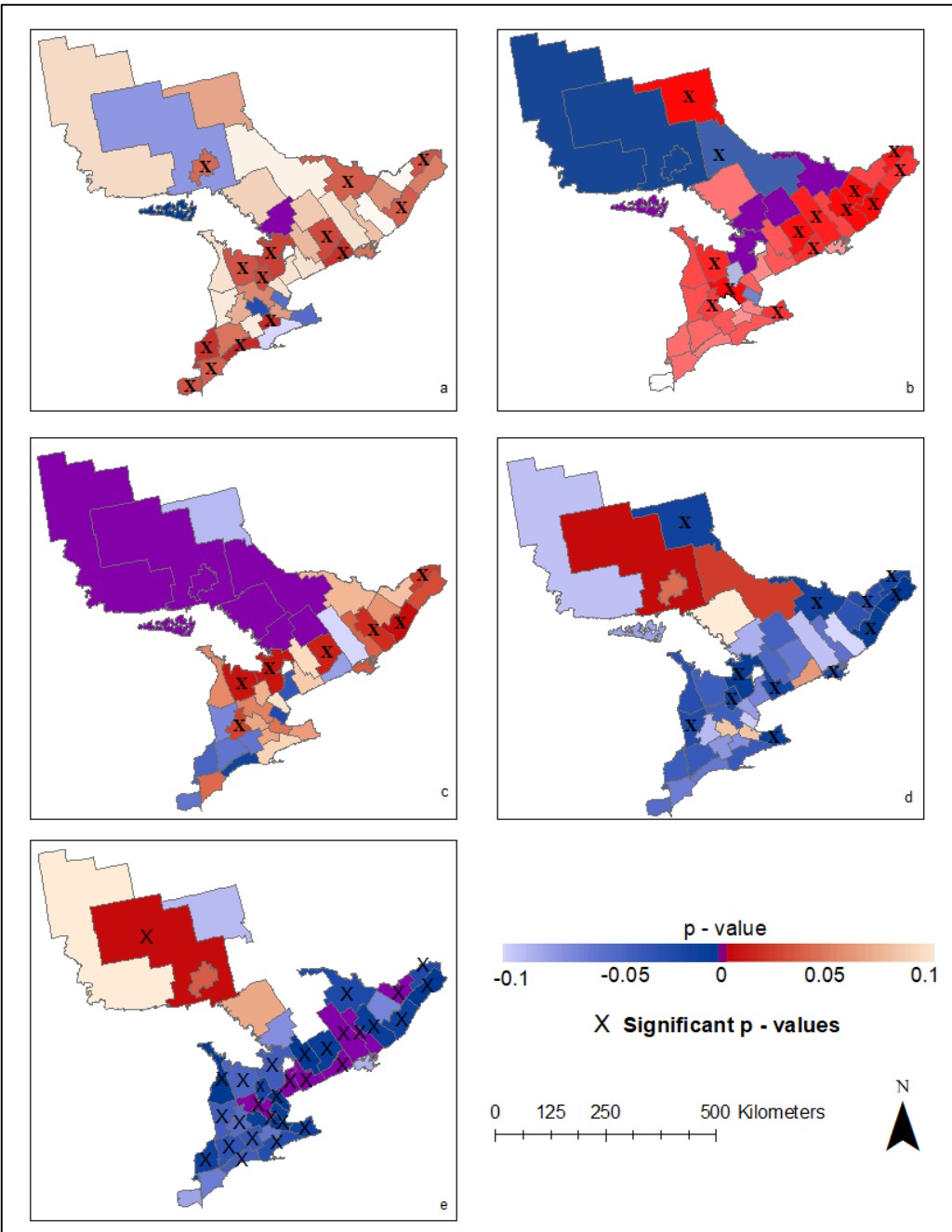

**Figure 4.** Long-term harvest trends in (**a**) corn, (**b**) soybean, (**c**) winter wheat, (**d**) hay, and (**e**) other field crops on the county level across Southern Ontario from 2004 to 2018.

Concurrent with soybean trends, corn production is increasing across many Southern Ontario's counties, with 14 counties exhibiting statistically significant increases in southwestern Ontario. This is observable, particularly in the north and northeast of Southern Ontario. The prominence of corn production in southwestern Ontario has been a consistent trend since 1961, as this area adopted United States Corn Belt agricultural systems [30]. Interestingly, southwestern Ontario appears to have a decline of winter wheat and no statistically significant increase in soybean production, suggesting that southwestern Ontario may not be adopting corn–soybean–winter wheat rotations. Alongside soybean, corn and winter wheat are increasing in production in the north and northeastern counties, indicating that corn–soybean–winter wheat rotations may be occurring in these regions.

*3.2. Cover Crop Adoption in Corn and Soybean Rotations across Southern Ontario*

The resilience of cropping systems under environmental stressors is of growing concern across many agricultural regions, as agrobiodiversity within crop regions is decreasing. This section presents an analysis of AAFC ACI data, identifying the extent to which cover crops are adopted into corn and soybean cropping systems. Figure 5 presents a quantitative analysis of the percent coverage of 28 rotation codes (Table 1). This perspective reveals the dominance of rotation code 10, occupying 2724 km² and approximately 25.6% of the total agricultural land analyzed in this study (Figure 5; Figure 6). Rotation 10 represents a two-crop rotation system of three years' corn and three years' soybean (3:0:3). The dominance of rotation 10 is an indication that most corn and soybean producers did not incorporate cover crops (here classified as winter wheat, rye, or oats) into the rotation over the past six years. Following rotation code 10, rotation codes 15, 6, and 13 are the second, third, and fourth most prominent rotation systems, respectively. As referred to in Table 1, rotations 6 and 15 represent systems exclusive to corn–soybean rotations; most notably, rotation code 13 represents equal frequencies of corn, soybean, and cover crops—this pattern is consistent with what this study deems a more optimal rotation for obtaining crop diversity. Spatial distributions of rotation practices are presented in Figure 6; counties dominant with rotation codes 10, 15, and 6 are aggregated around south and southwestern Ontario and become less prominent in counties progressing north. In comparison, rotation code 13 appears to be more prominent in counties progressing northward, indicating a greater adoption of cover crops in corn–soybean rotations in the north.

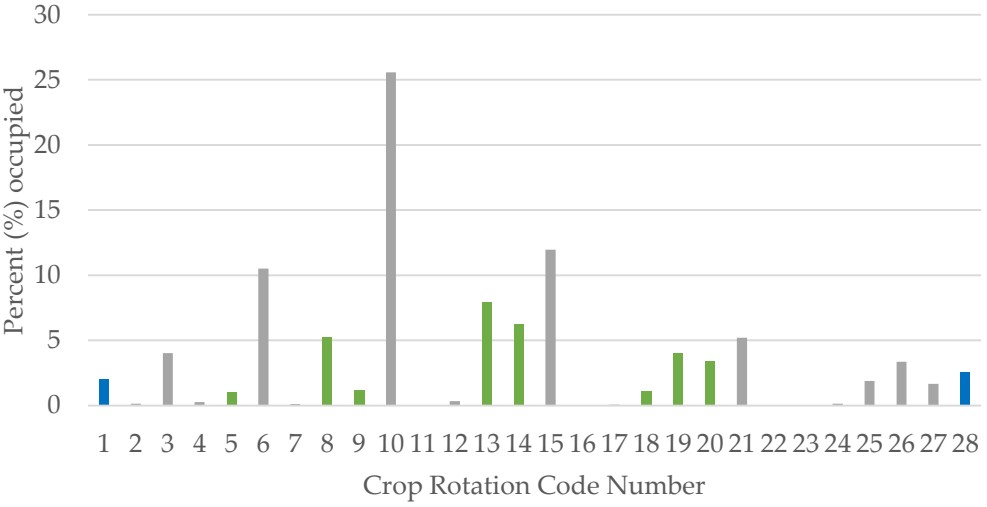

**Figure 5.** Bar chart illustrating the percent of area occupied by the crop rotation types (refer to Table 1 for meaning). Bars colored green are indicative of a three-crop rotation, two-crop rotations are in gray, and monoculture are in blue colors.

The study's second analysis of cover crop adoption aggregates the rotations that share the same practice (Table 2)—monoculture, two-crop rotations, three-crop rotations. This perspective effectively reveals the extent to which the selected cover crops are incorporated into corn and soybean rotations. The incorporation of cover crops may still be an underutilized soil conservation method in Southern Ontario. This study reveals that, in general, corn and soybean cropping systems are not incorporating cover crops regularly into rotations. Instead, this study reveals that producers in corn and soybean production are more likely to adopt two-crop rotation cropping systems, representing 64.8% of the occupied area (Figure 7).

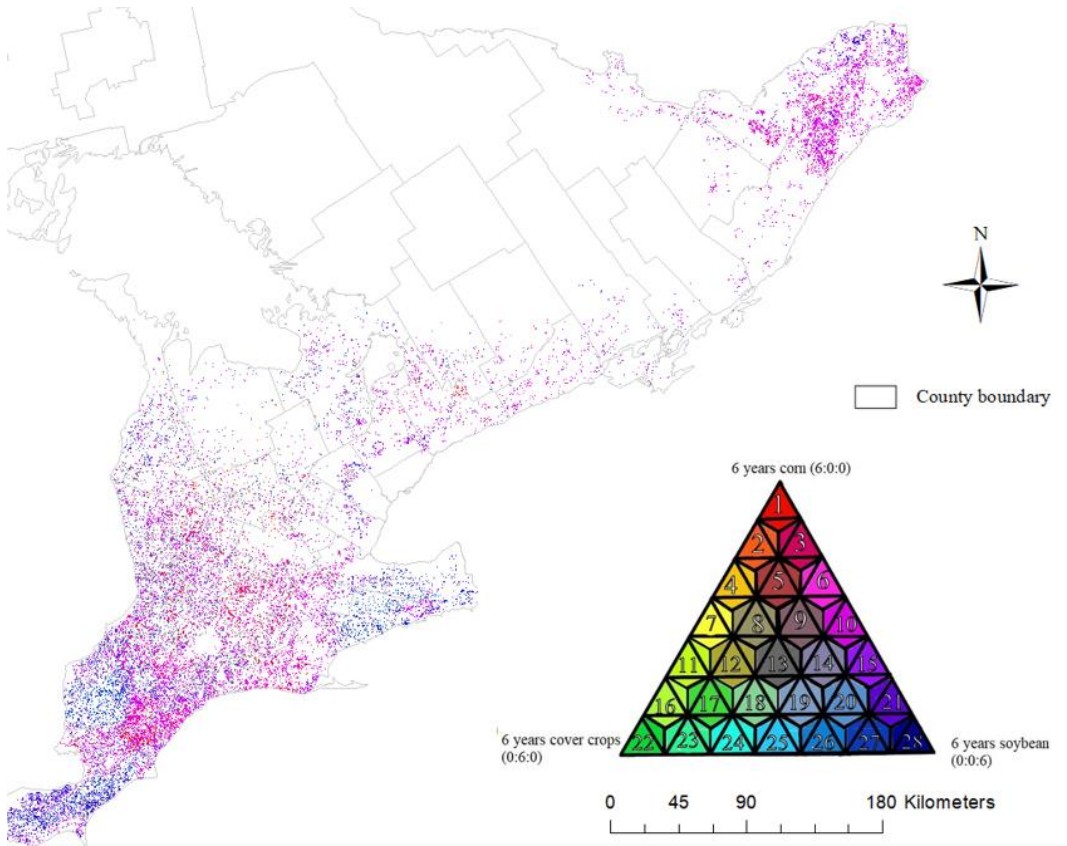

**Figure 6.** Areas of crop rotation based on rotation types (1–28) considering corn, soybean, and/or winter wheat in Southern Ontario from 2013 to 2018.

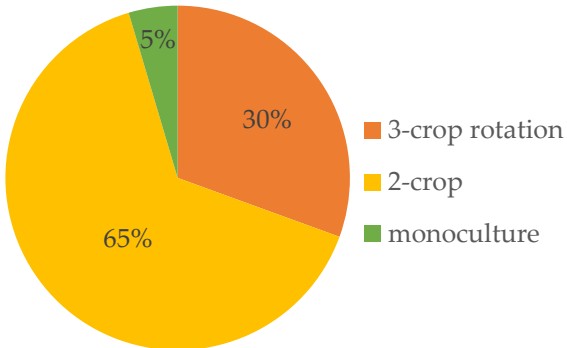

**Figure 7.** Percent of the area occupied by three-crop, two-crop, and monoculture, based on the ratio of corn, soybean, and cover crops grown.

The intermediate rotations are consistent with the findings from Figure 5, as rotation code 4 (Figure 8, Table 2), much like rotation code 10 (Figure 5, Table 1), represents a two-crop rotation. Code 4, representing 27.6% of rotations in Figure 8, contains rotations 7, 10, and 25 (Figure 5); however, rotation code 7 and 25 occupancies is limited, representing only 0.43 and 6.8%, respectively, while rotation code 10 represents 92.8%. The limited occurrence of a two-crop rotation of corn cover crops or soybean cover crops suggests that most of the land occupied by rotation 4 (Figure 8) amounts to equal two-crop rotation between corn and soybean (rotation code 10). Although two-crop rotations prevail, diversification of corn–soybean rotations with cover crops represents a substantial portion of corn and soybean cropping systems, representing 30.6% of the analyzed areas (Figure 7).

Serendipitously, monoculture has the lowest adoption rate in corn and soybean production, representing only 4.6% of the analyzed area (Figure 7).

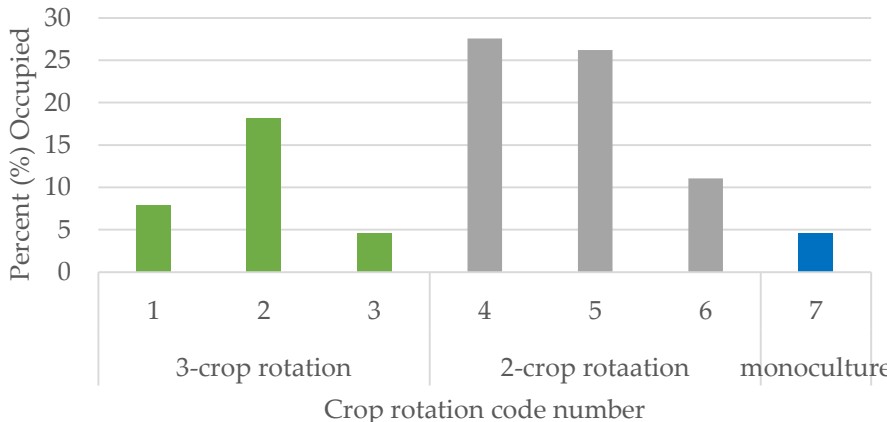

**Figure 8.** Percent of the area occupied by rotation types as per Table 2, representing three-crop, two-crop, and monoculture practices of corn, cover crops, and soybean.

The spatial distribution of monoculture, two-crop rotations, and three-crop rotations vary over the study site. Concurrent with county trends in Figure 4, counties located in southwestern Ontario are aggregated with monoculture and two-crop rotations (Figure 9), given the saturation of red symbology in this area. The predominance of two-crop rotations aggregated in southwestern Ontario supports the findings in Figure 4, which illustrate the decline of winter wheat in this area, suggesting southwestern Ontario is prominently growing corn–soybean rotations in this area. Proceeding north of the study site, Figure 9 shows an aggregation of purple representing the incorporation of three-crop rotations of corn, soybean, and cover crops. The northward adoption of a three-crop rotation supports Figure 4 findings, as counties in this region are increasing in corn, soybean, and cover crops rotation.

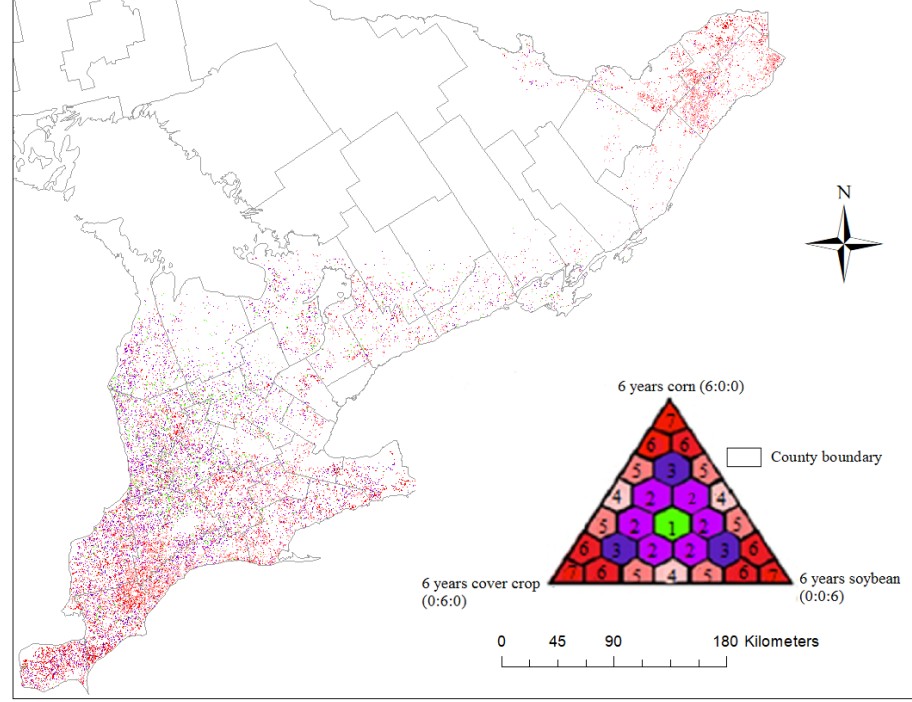

**Figure 9.** Areas of crop rotation for seven rotation types, considering corn, soybean, and or winter wheat in Southern Ontario from 2013 to 2018.

There is little evidence of true monoculture systems, as continuous soybeans represent 2.5% (rotation code 28), and continuous corn (rotation code 1) represents 2.1% of the analyzed operations (Figure 5; Figure 7). The previous analysis of OMAFRA's Field Crop Data from 1981 to 2018 suggests a significant increase in soybean in Southern Ontario. An increase in soybean across Southern Ontario is consistent with O'Neal et al. [31] study that modeled future shifts from corn and wheat to soybean in Midwestern United States. O'Neal et al. [31] suggest a regime shift toward continuous soybean production will ultimately increase soil erosion and, in turn, decrease productivity. The projections discussed by O'Neal et al. [31] are concerning for soil productivity in Southern Ontario if cropping systems divert from agrobiodiversity and cover crops toward soybean monoculture.

## 4. Conclusions

Climate change and conventional agricultural practices, especially monoculture, are concerning for soil productivity, and more diverse rotations offer yield stability [7] and improve the accumulation of soil organic carbon, particularly when combined with no-till approaches [19]. While these practices are desirable socially and environmentally, there may be some reluctance amongst producers to adopt such practices due to the costs that concentrate at the farm level [32,33]. Research is somewhat limited on the extent to which this recommendation of incorporating more diverse rotations, including cover crops into traditional corn and soybean rotations, has been adopted at provincial scale and at the farm level within Southern Ontario. Using crop survey data and high-resolution crop mapping products, this work has assessed the uptake of cover crops (here defined as winter wheat or cereals) put into rotation within corn and soybean operations across Ontario.

Using survey data from 1981 to 2018 at the county level, trends in the production of various crops are analyzed. The findings, consistent with previous literature, particularly within the Corn Belt of the United States (e.g. [28] and Ontario [7,30], suggest a reduction in crop diversity with a continued focus on corn and soybean cropping systems. In particular, a trend analysis suggests that soybean, corn, and winter wheat have increased across several counties, while hay and other field crops are declining. This study reveals the dominance of two crops in particular, corn and soybean, often grown in rotation. Detailed farm level analysis using an annual crop inventory suggests that substantial portion (30%) of corn and soybean operations observed in this study incorporated more diverse rotations that featured cover crops (here defined as winter wheat or other winter cereals) into a three-crop system. Spatially, it is found that operations in southwestern Ontario are predominantly in corn–soybean rotations, incorporating fewer crop covers into rotations. This study suggests that further efforts should extend to these regions for incorporating more diverse cover crops, specifically in corn and soybean operations within southwestern Ontario. However, significant progress is occurring in Ontario with the application of winter cover crops (generally mixtures of leguminous and non-leguminous covers) either inter-seeded in a growing crop or planted at harvest. These types of covers were not tracked in the agricultural crop inventory, therefore, trends on the use of these covers should be implemented in future work.

The findings in this paper support broad characterizations described across agricultural regions throughout North America, revealing a decline in crop rotational diversity. They highlight a shift in agrobiodiversity to less diverse cropping systems. While the findings may be used to identify where diversity enhancement programs within Southern Ontario may be targeted, this study opens the door for future studies that may reveal other vital variables that are influencing producers' decisions in implementing OMAFRA's recommendations. Further development of techniques and datasets (e.g. [24] for the tracking of the non-growing season agriculture land cover practices, is also recommended.

**Author Contributions:** Conceptualization, K.S., K.B.K. and A.B.; methodology, K.S., K.B.K. and A.B.; software, K.S., K.B.K. and A.B.; validation, K.S.; formal analysis, K.S.; investigation, K.S., K.B.K. and A.B.; resources, K.B.K. and A.B.; data curation, K.S.; writing—original draft preparation, K.S., K.B.K. and A.B.; writing—review and editing, K.S., K.B.K. and A.B.; visualization, K.S. and K.B.K.; supervision, K.B.K. and A.B.; project administration, K.B.K. and A.B.; funding acquisition, K.B.K. and A.B. All authors have read and agreed to the published version of the manuscript.

**Funding:** This work was supported by the Food from Thought Program at the University of Guelph, funded by the Canada First Research Excellence Fund.

**Data Availability Statement:** All relevant data are within the paper.

**Acknowledgments:** The authors would like to acknowledge the early contribution of Justin Adams for development of some scripts for corn and soybean mapping.

**Conflicts of Interest:** The authors declare no conflict of interest.

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
