# Peer review of "Evaluation of Agrobiodiversity and Cover Crop Adoption in Southern Ontario Field Crops"

_agronomy, doi:10.3390/agronomy12020415_

Round 1
Reviewer 1 Report
The manuscript results could be supported with the benifit : cost ratio of cropping systems and feasiblity of economically best cover crop.
then conclude the best cover crop for inclusion in cropping system
Author Response
Reviewer #1
The manuscript results could be supported with the benifit : cost ratio of cropping systems and feasiblity of economically best cover crop.
then conclude the best cover crop for inclusion in cropping system
- First of all, we would like to thank you for the endorsement of our paper. We greatly appreciate your valuable time in reviewing our paper. We agree with your remark that it would be great if this manuscript also consists of the benefit-cost analysis for different cropping rotations and the conclusion of an economically feasible cropping system for the study area. However, performing a benefit-cost ratio for different cropping systems is beyond the scope of this paper as this paper was based on the spatial analysis of historical agricultural land use data from remote sensing images.
Reviewer 2 Report
Thank you for the opportunity to review this publication. Authors present interesting trends in cover crop adoption and agrobiodiversity in Ontario, Canada. However, author's make several conclusions that are not supported by their current methodology. Also, given that 28 codes are used in one analysis and 7 codes in another it would be helpful to the reader to refer to these by crop rotation in the text. For example lines 250-251 could read, "This perspective reveals the dominance of a three year corn followed by a three year soybean rotation (rotation 10). Please see individual comments below.
Line 32: change to "corn, wheat, and soy..."
Line 77: Define "OMAFRA" at first mention.
Lines 83-84: Please explain these rotations better. Is it soybean followed by a cover crop followed by corn (Soybean/Cover Crop/Corn) and corn followed by a cover crop followed by soybean (Corn/Cover Crop/Soybean)? Should line 84 read "continuous corn, continuous soybean..."
Equation 1: Describe equation in detail, what does each value represent.
Figure 2. Consider changing the colors or outlining the numbers in black. White numbers are difficult to see on a yellow background. Suggest 1 figure combining Table S1 with figure 2a and a separate figure with Table S2 with figure 2b. Space needed between "A" and "28" and "B" and 7.
Line 192: Delete "More significantly", figure three does not appear to show any true statistical comparisons.
Figure 4: This figure would we better interpreted as a significant increase or decrease in cropping systems. Pick one p-value. It appears multiple p-values were used which can influence interpretation.
Line 250: "2,724"
Line 256: "Table S1". This table is key to interpreting the results and should not be supplementary material.
Line 258: Delete "Table S1" as it is previously referred to in the same sentence.
Line 260: Shouldn't this refer to figure 6.
Line 262-264: This trend is not obvious on the figure provided.
Figure 5: Statistical values are needed. Did author's conduct any means comparisons tests to distinguish crop rotation codes. "Colored" is spelled differently here, please review author guidelines and be consistent with spelling.
Line 272: Not sure which analysis is being referred to.
Figure 8. See comments for Figure 5.
Line 325: Author's present no data to support how climate change and monocultures influence soil productivity. Including weather data for the study period would improve the discussion. No till agriculture can also help to conserve soil without a cover crop, however, authors do not factor in tillage systems in their conclusions.
Line 340: delete "significantly"
Line 341: replace "chapter" with "study"
Tables S1 and S2: An explanation of what the numbers represent needs to be included in the table.
Author Response
Reviewer #2
Thank you for the opportunity to review this publication. Authors present interesting trends in cover crop adoption and agrobiodiversity in Ontario, Canada. However, author's make several conclusions that are not supported by their current methodology. Also, given that 28 codes are used in one analysis and 7 codes in another it would be helpful to the reader to refer to these by crop rotation in the text. For example lines 250-251 could read, "This perspective reveals the dominance of a three year corn followed by a three year soybean rotation (rotation 10). Please see individual comments below.
- We would like to thank you for the endorsement of our paper. We greatly appreciate your valuable time in reviewing our paper. We thank you for your suggestions for using the phrase “rotation” in front of the code. In the revised manuscript we implemented your suggestion by adding the phrase “rotation” in front of the code at all the relevant places wherever it was appropriate.
Line 32: change to "corn, wheat, and soy..."
- We have changed it.
Line 77: Define "OMAFRA" at first mention.
- We have elaborated the full form of OMAFRA as the Ontario Ministry of Agriculture, Food and Rural Affairs (OMAFRA) in the revised manuscript.
Lines 83-84: Please explain these rotations better. Is it soybean followed by a cover crop followed by corn (Soybean/Cover Crop/Corn) and corn followed by a cover crop followed by soybean (Corn/Cover Crop/Soybean)? Should line 84 read "continuous corn, continuous soybean..."
- Thanks for this great suggestion. We have explained them in the revised manuscript as per your suggestions.
Equation 1: Describe equation in detail, what does each value represent.
- In the revised manuscript, all components of the equation were described and elaborated.
Figure 2. Consider changing the colors or outlining the numbers in black. White numbers are difficult to see on a yellow background. Suggest 1 figure combining Table S1 with figure 2a and a separate figure with Table S2 with figure 2b. Space needed between "A" and "28" and "B" and 7.
- Again thanks for this great suggestion. We have incorporated your suggestions and revised the figure and text.
Line 192: Delete "More significantly", figure three does not appear to show any true statistical comparisons.
- The phrase “more significantly” has been removed in the revised manuscript.
Figure 4: This figure would we better interpreted as a significant increase or decrease in cropping systems. Pick one p-value. It appears multiple p-values were used which can influence interpretation.
- In fact, we think the negative and positive p-values are clearly demonstrated what exactly you are suggesting/asking as a significant increase or decrease in cropping systems. So we believe instead of picking one p-value the figure as its current state is more meaningful. So we have decided to present this figure in its current form.
Line 250: "2,724"
- This is 2,724km2
Line 256: "Table S1". This table is key to interpreting the results and should not be supplementary material.
- We like your suggestion in the revised manuscript we have presented both Table S1 and Table S2 in the main text as Table1 and Table 2.
Line 258: Delete "Table S1" as it is previously referred to in the same sentence.
- Deleted
Line 260: Shouldn't this refer to figure 6.
- Yes, this refers to both Figure 5 and Figure 6. In the revised manuscript we have mentioned it accordingly.
Line 262-264: This trend is not obvious on the figure provided.
- We have rewritten this in the revised manuscript.
Figure 5: Statistical values are needed. Did author's conduct any means comparisons tests to distinguish crop rotation codes. "Colored" is spelled differently here, please review author guidelines and be consistent with spelling.
- We are sorry to say that no statistical test was performed to compare different types of crop rotations. The main intention of presenting this bar chat is to demonstrate the percent areas under each type of crop rotation.
Line 272: Not sure which analysis is being referred to.
- Thank you for pointing this out. In fact, this refers to figure 7, pie chart. We have elaborated this in the revised manuscript.
Figure 8. See comments for Figure 5.
- As we mentioned above to respond to your comment for figure 5, we are sorry to say that no statistical test was performed to compare different types of crop rotations. The main intention of presenting this bar chat is to demonstrate the percent areas under each type of crop rotation. This figure is for seven crop rotations types and in figure 5 for 28 crop rotation types.
Line 325: Author's present no data to support how climate change and monocultures influence soil productivity. Including weather data for the study period would improve the discussion. No till agriculture can also help to conserve soil without a cover crop, however, authors do not factor in tillage systems in their conclusions.
- Thanks for your remarks. We have rewritten the opening part of the conclusion as “Climate change and conventional agricultural practices, especially monoculture, is concerning to soil productivity and that more diverse rotations offers yield stability (e.g. Gaudin et al. 2015b) and improve the accumulation of soil organic carbon particularly when combined with no-till approaches (Laamrani et al. 2020a).
Line 340: delete "significantly"
- Deleted
Line 341: replace "chapter" with "study"
- Replaced
Tables S1 and S2: An explanation of what the numbers represent needs to be included in the table.
- Based on your suggestion both supplementary tables were incorporated into the main text as Table 1 and Table 2. An explanation was added both in table captions and in the column heading.
Reviewer 3 Report
- The summary must be populated with numbers, with percentages, with concrete data. The paper is quite rich in data to meet this requirement.
- I suggest adding to the KEYWORDS and the phrase: ”cover crops”.
- At the end of the INTRODUCTION chapter, the working hypotheses must be stated clearly and concisely.
- The discussion part is quite poor. The authors limited themselves to bibliographic references to Ontario and possibly the USA. In order to increase the relevance and value of research, I propose to extend the comparative studies with research carried out in Europe and other parts of the world, where cover crops are practiced for the two target species (maize and soybeans).
- The conclusions are too rich. The conclusions are not a recapitulation of the results but represent strong statements based on scientific arguments and their logical interpretation. I suggest that the information be concentrated in a smaller space and should show the novelty of the results and the extent to which the results obtained met the hypotheses in the introduction.
Author Response
Reviewer #3
- The summary must be populated with numbers, with percentages, with concrete data. The paper is quite rich in data to meet this requirement.
- Thank you very much for your great suggestion. However, as we mentioned in our paper the main objective of our paper is to explore the frequency of which corn and soybean productions have incorporated with what crop crops in the six years of the study period. In a spatial context, we observed that Southern Ontario is shifting from systems characterized by higher agrobiodiversity to systems dominated by corn, soybean, and hay. However, most of the current corn and soybean operations are not incorporating cover crops into the rotation, particularly in southwestern Ontario. We have mentioned this in our abstract.
- I suggest adding to the KEYWORDS and the phrase: ”cover crops”.
o Great suggestion. We have added the phrase “cover crops” as one additional keyword
- At the end of the INTRODUCTION chapter, the working hypotheses must be stated clearly and concisely.
- Thanks for these comments. We believe paper can be written either with a working hypothesis or for specific research questions and/or objectives. We have clearly stated at the end of the third paragraph of the introduction section that the objective of this paper is to explore which cover crops are adopted in corn and soybean operations, as well as trends in cover crop spatial adoption across Southern Ontario in Canada?
- The discussion part is quite poor. The authors limited themselves to bibliographic references to Ontario and possibly the USA. In order to increase the relevance and value of research, I propose to extend the comparative studies with research carried out in Europe and other parts of the world, where cover crops are practiced for the two target species (maize and soybeans).
- In the revised manuscript, we have elaborated the discussion by incorporating a couple of bibliographic references (please see the new references that were added in the revised manuscript).
- The conclusions are too rich. The conclusions are not a recapitulation of the results but represent strong statements based on scientific arguments and their logical interpretation. I suggest that the information be concentrated in a smaller space and should show the novelty of the results and the extent to which the results obtained met the hypotheses in the introduction.
- Thanks for your suggestion. We have rewritten the conclusion section. It now reads as: Climate change and conventional agricultural practices, especially monoculture, is concerning to soil productivity and that more diverse rotations offer yield stability (e.g. Gaudin et al. 2015b) and improve the accumulation of soil organic carbon particularly when combined with no-till approaches (Laamrani et al. 2020a). While these practices are desirable socially and environmentally, there may be some reluctance among producers to adopt such practices due to the costs that concentrate at the farm level (Knowler & Bradshaw, 2007). Research is somewhat limited on the extent to which this recommendation of incorporating more diverse rotations including cover crops into traditional corn and soybean rotations has been adopted at provincial scale and at the farm level within Southern Ontario. Using crop survey data, and high-resolution crop mapping products, this work has assessed the uptake of cover crops (here defined as winter wheat or cereals) put into rotation within corn and soybean operations across Ontario.
Using survey data from 1981-2018 at the county level, trends in the production of various crops are analyzed. The findings, consistent with previous literature, particularly within the Corn Belt of the United States (e.g. Wright and Wimberly, 2013) and Ontario (Hume & Pearson, 2011; Gaudin et al. 2015b), suggest a reduction in crop diversity with a continued focus on corn and soybean cropping systems. In particular, a trend analysis suggests that soybean, corn, and winter wheat have increased across several counties while hay and other field crops are declining. This study reveals the dominance of 2 crops, in particular, corn and soybean were often grown in rotation. Detailed farm-level analysis using an annual crop inventory suggests that a substantial portion (30%) of corn and soybean operations observed in this study incorporated more diverse rotations that featured a cover crop (here defined as winter wheat or other winter cereals) into a 3-crop system. Spatially, it is found that operations in southwestern Ontario are predominately in corn-soybean rotations – incorporating fewer crop covers crops into rotations. This study suggests that further efforts should extend to these regions for incorporating more diverse adopt cover crops, specifically in corn and soybean operations within southwestern Ontario. However, significant progress is occurring in Ontario with the application of over winter cover crops (generally mixtures of leguminous and non-leguminous covers) either inter-seeded in a growing crop or planted at harvest, these types of covers were not tracked in the agricultural crop inventory, therefore, trends on the use of these covers should be implemented in future work.
The findings in this paper support broad characterizations described across agricultural regions throughout North America, revealing a decline in crop rotational diversity. It highlights a shift in agrobiodiversity to less diverse cropping systems. While the findings may be used to identify where diversity enhancement programs within Southern Ontario may be targeted, this study opens the door for future studies that may reveal other vital variables that are influencing producers' decisions for implementing OMAFRA's recommendations. Further development of techniques and data sets (e.g. Laamrani et al. 2020b) for the tracking of the non-growing season agriculture land cover practices is also recommended.